# Structural and Biochemical Analyses of the Butanol Dehydrogenase from *Fusobacterium nucleatum*

**DOI:** 10.3390/ijms24032994

**Published:** 2023-02-03

**Authors:** Xue Bai, Jing Lan, Shanru He, Tingting Bu, Jie Zhang, Lulu Wang, Xiaoling Jin, Yuanchao Mao, Wanting Guan, Liying Zhang, Ming Lu, Hailong Piao, Inseong Jo, Chunshan Quan, Ki Hyun Nam, Yongbin Xu

**Affiliations:** 1Department of Bioengineering, College of Life Science, Dalian Minzu University, Dalian 116600, China; 2Key Laboratory of Biotechnology and Bioresources Utilization of Ministry of Education, College of Life Science, Dalian Minzu University, Dalian 116600, China; 3School of Life Science and Biotechnology, Dalian University of Technology, No. 2 Linggong Road, Dalian 116024, China; 4Shandong Provincial Key Laboratory of Energy Genetics, Key Laboratory of Biofuel, Qingdao Institute of Bioenergy and Bioprocess Technology, Chinese Academy of Sciences, Qingdao 266101, China; 5CAS Key Laboratory of Separation Science for Analytical Chemistry, Dalian Institute of Chemical Physics, Chinese Academy of Sciences, Dalian 116023, China; 6Infectious Diseases Therapeutic Research Center, Korea Research Institute of Chemical Technology, Daejeon 34114, Republic of Korea; 7Department of Life Science, Pohang University of Science and Technology, Pohang 35398, Republic of Korea; 8POSTECH Biotech Center, Pohang University of Science and Technology, Pohang 35398, Republic of Korea

**Keywords:** *Fusobacterium nucleatum*, butanol dehydrogenase (BDH), YqdH, NAD(P)H, metal ion

## Abstract

Butanol dehydrogenase (BDH) plays a significant role in the biosynthesis of butanol in bacteria by catalyzing butanal conversion to butanol at the expense of the NAD(P)H cofactor. BDH is an attractive enzyme for industrial application in butanol production; however, its molecular function remains largely uncharacterized. In this study, we found that *Fusobacterium nucleatum* YqdH (FnYqdH) converts aldehyde into alcohol by utilizing NAD(P)H, with broad substrate specificity toward aldehydes but not alcohols. An in vitro metal ion substitution experiment showed that FnYqdH has higher enzyme activity in the presence of Co^2+^. Crystal structures of FnYqdH, in its apo and complexed forms (with NAD and Co^2+^), were determined at 1.98 and 2.72 Å resolution, respectively. The crystal structure of apo- and cofactor-binding states of FnYqdH showed an open conformation between the nucleotide binding and catalytic domain. Key residues involved in the catalytic and cofactor-binding sites of FnYqdH were identified by mutagenesis and microscale thermophoresis assays. The structural conformation and preferred optimal metal ion of FnYqdH differed from that of TmBDH (homolog protein of FnYqdH). Overall, we proposed an alternative model for putative proton relay in FnYqdH, thereby providing better insight into the molecular function of BDH.

## 1. Introduction

Butyrate biosynthesis by the gut microbiota plays a critical role in maintaining health and, if altered, can lead to pathological conditions, such as inflammatory bowel diseases [1], ulcerative colitis, and type II diabetes [2,3]. Butyrate, the preferred energy source for colonocytes, is protective in colorectal cancer [4]. It is generated from carbohydrates via the glycolysis of two acetyl-CoA molecules yielding acetoacetyl-CoA, followed by stepwise reduction to butyryl-CoA [3]. Four pathways with different initial substrates (pyruvate, 4-aminobutyrate, glutarate, and lysine) are known for butyrate formation [5,6]. Each of these four pathways uses the butyryl-CoA dehydrogenase-electron transferring flavoprotein complex (Bcd-Etf) to catalyze the conversion of crotonyl-CoA to butyryl-CoA [7]. Eventually, the final butyrate production is catalyzed by butyryl-CoA transferase (BCoAT) or butyrate kinase (Buk) [8,9].

*Fusobacterium nucleatum* is a gram-negative anaerobic bacterium with species-specific reservoirs in the human oral cavity or gastrointestinal tract [10,11,12]. Recent accumulating evidence has shown that some *Fusobacterium*, especially *F. nucleatum*, are related to colorectal cancer progression [10,13]. In the cancer microenvironment, a decrease in *F. nucleatum* butyrate metabolism synergistically promotes inflammatory responses [13,14]. Whether there is a bidirectional effect between an *F. nucleatum*-mediated decrease of butyrate production and tumor development remains to be studied. Butyrate can be biologically reduced to butanol, during the solventogenic phase, via CoA transferase (CTF), aldehyde/alcohol dehydrogenase (AADH), and butanol dehydrogenase (BDH) (Figure 1) [15,16].

BDH plays a significant role in butanol production in many microorganisms by catalyzing butanal conversion to butanol at the expense of the NAD(P)H cofactor [17]. Homologs of BDH have been found in other anaerobic bacteria, including *Clostridium acetobutylicum* [18] and *Clostridium beijerinkii* [19], and they are involved in the n-butanol biosynthetic pathway. *F. nucleatum* possesses a putative *yqdH* gene that encodes the BDH protein and plays an essential role in butanol biosynthesis and butyrate metabolism [20]. Consequently, *F. nucleatum* YqdH (named FnYqdH) is likely to be a potential drug target for developing new types of antibiotics. However, the structure and function of FnYqdH have not been studied yet, and the molecular mechanism of butanol synthesis in anaerobic bacteria remains to be explored in detail.

In this study, we found that FnYqdH exhibited dual cofactor specificity for nicotinamide adenine dinucleotide (NADH) and nicotinamide adenine dinucleotide phosphate (NADPH) and has butanol dehydrogenase activity for various aldehydes. In addition, four metal ions, namely Co^2+^, Ni^2+^, Zn^2+^, and Fe^3+^, significantly enhanced the activity of metal-free FnYqdH. Moreover, FnYqdH showed broad substrate specificity toward aldehydes. To gain structural insights into the catalytic mechanism, along with substrate and cofactor specificities of FnYqdH, we determined the crystal structures of YqdH from *F. nucleatum* in its apo and complexed forms (with NADH and Co^2+^). We found several residues involved in the coenzyme- and metal ion-binding of FnYqdH. Our study on FnYqdH could offer valuable clues for a better understanding of the molecular functions of BDH, which is a potential antibacterial target for *F. nucleatum*-related disease.

## 2. Results

### 2.1. Biochemical Analysis of FnYqdH

BDHs are classified as BDHA and BDHB based on their use of NADH and NADPH as cofactors, respectively [21]. In the GenBank, FnYqdH (accession number: AAL95608.1) has been annotated as NADH-dependent BDHA, indicating that it catalyzes butanal conversion to butanol using the NADH cofactor. BDHs essentially require a metal ion for substrate recognition during a dehydrogenase reaction, and they play a pivotal role in enzymatic activity [22,23,24]. To investigate the cofactor specificity and examine the effect of a metal ion on FnYqdH, the enzyme activities of native FnYqdH, EDTA-FnYqdH (metal-free; the metal of FnYqdH is removed by incubation with EDTA, a chelating agent), and that with the addition of Co^2+^ (native-Co^2+^ and EDTA-Co^2+^, respectively) were measured using butanal as the substrate, with NADH or NADPH as cofactors, respectively. EDTA-FnYqdH had almost no BDH activity in the presence of NADH or NADPH, indicating that the metal ions are essential. Meanwhile, when adding Co^2+^ to EDTA-FnYqdH, the enzyme activity was recovered. When adding Co^2+^ to native FnYqdH, the enzyme activity was enhanced by 50–60% in the presence of NADH or NADPH, indicating that the functional metal ion sites were not fully occupied on purified FnYqdH. Notably, FnYqdH showed BDH activity with both NADH and NADPH (Figure 2a). In the presence of 4.5 μM FnYqdH, the consumption of NADH and NADPH was 417 μM and 357 μM, respectively, (Figure 2a and Appendix A), indicating the ability of FnYqdH to use both NADH and NADPH as cofactors. Furthermore, the apparent *K*_m_ and *V*_max_ values were measured with a purified protein. The results showed that NADH is preferred over NADPH in terms of FnYqdH activity (Appendix A). Subsequently, BDH activity was measured after incubating metal-free FnYqdH with metal ions (Mg^2+^, Co^2+^, Ni^2+^, Zn^2+^, and Fe^3+^). Since Fe^2+^ is easily oxidized to Fe^3+^ at pH 6.0 [25], butanol dehydrogenase activity was monitored only in the presence of Fe^3+^ to avoid possible experimental errors when Fe^2+^ was used. The highest enzymatic activity of FnYqdH was observed in the presence of Co^2+^; this activity was absent in the presence of Mg^2+^. The overall BDH activity trend was as follows: Ni^2+^ > Zn^2+^ > Fe^3+^, with relative activities of 93.7%, 85.2%, and 44.2%, respectively (Figure 2b).

To clarify the substrate specificity of FnYqdH, its activity was measured for ten different aldehydes (methanal, ethanal, propanal, butanal, pentanal, hexanal, heptanal, octanal, nonanal, and decanal). FnYqdH showed high activity for propanal (substrate degradation rate: 68%), butanal (83%), pentanal (96%), hexanal (92%), heptanal (100%), and octanal (77%) (Figure 2c); however, it showed very low enzyme activity for methanal (18%), ethanal (51%), nonanal (31%), and decanal (20%) (Figure 2c). Collectively, the results suggested that FnYqdH can convert aldehydes to alcohols, although the activity differed according to the length of the aliphatic chain—aldehydes with 3–8 carbon atoms in the aliphatic chain were preferred.

Protein BLAST analysis of FnYqdH revealed its high sequence similarity with several alcohol dehydrogenases (ADHs) from *Abyssisolibacter fermentans*(Accession code: WP_066501503.1, sequence identity: 57%), *Clostridium bovifaecis* (QGU94363.1, 55%), *Defluviitalea raffinosedens* (WSLF01000004, 54%), *Epulopiscium* sp. (NLK98329.1, 54%), and *Clostridiaceae bacterium* (NLY43117.1, 52%) (Appendix A). ADHs are ubiquitous enzymes that catalyze the oxidation of alcohols and reduction of aldehydes in the presence of NAD(P)^+^ and NAD(P)H, respectively [26,27]. To determine the potential ADH activity of FnYqdH, the oxidase activity of FnYqdH was screened for ten different alcohols (methanol, ethanol, propanol, butanol, pentanol, hexanol, heptanol, octanol, nonanol, and decanol) in the presence of NAD^+^. However, FnYqdH showed no significant enzymatic activity against any alcohol (Figure 2d), indicating that FnYqdH only has reductase activity for aldehydes, contrary to ubiquitous ADHs.

### 2.2. Overall Structure of FnYqdH

To better understand the molecular function of FnYqdH, we determined the crystal structure of FnYqdH using purified recombinant FnYqdH at 1.98 Å. However, metal ions and NAD(P)H were not observed in the electron-density map (named FnYqdH-apo). To identify the cofactor-binding form of FnYqdH, FnYqdH crystals were soaked for 1 min in a reservoir solution supplemented with NADH and Co^2+^. After, we determined the crystal structure of FnYqdH complexed with NADH and Co^2+^ at 2.72 Å (named FnYqdH-NAD-Co^2+^) (Table 1). The FnYqdH crystals belonged to the orthorhombic space group I222, containing one molecule in the asymmetric unit. The R_work_/R_free_ of the final model of FnYqdH-apo and FnYqdH-NAD-Co^2+^ were 0.177/0.209 and 0.205/0.260, respectively.

The monomer FnYqdH structure consisted of an NAD-binding domain (NBD) and a catalytic domain (CD), connected by a 3_10_–helix ƞ3 (linker). Further, it has a deep cleft between NBD and CD (Figure 3a). The NBD (residues 1–180) had a pair of Rossmann-like α/β/α sandwich folds formed by six parallel β-sheets (β2 to β6 and β9) and surrounded by six α-helices (α1 to α6) (Figure 3b). The two anti-parallel strands, β7 and β8, formed a long β-hairpin and were flanked by two 3_10_-helices, ƞ1 and ƞ2. The CD (residues 190–385) contained nine α-helices and two 3_10_-helices, ƞ4 and ƞ5, thereby representing typical dehydrogenase synthase-like α domain topology (Figure 3b). B-factor analysis of FnYqdH revealed that the NBD region (B-factor: 35.00 Å^2^) was more flexible than the CD region (22.40 Å^2^) while mobile loops gathered on the surface (Appendix A). Compared to B-factors of the main chain (29.01 Å^2^), B-factor analysis further showed that most regions form a rigid structure, excluding the L4 (Lys32–Thr34, B-factor: 54.50 Å^2^), α2 (Glu45–Leu50, 51.70 Å^2^), L5 (Glu41–Gly44, 47.10 Å^2^), and L15 (Glu160–Lys163, 63.40 Å^2^). The conserved surface of FnYqdH exhibited the substrate-binding cleft as highly conserved, while other regions had no or relatively low amino acid conservation (Figure 3c). 

In the crystal structure, two molecules of FnYqdH formed a dimer in symmetric crystal packing (Figure 3d), which was consistent with the dimeric state of FnYqdH as evidenced by size-exclusion chromatography (Figure 3e). The dimeric formation of FnYqdH was stabilized via an anti-parallel β-sheet formed between the N-terminal β1 of one molecule and β2′ strands of the second (Figure 3d). Dimeric interface analysis revealed the hydrogen bonds between three residue pairs (Asn3-Lys16*, Phe4-Phe14*, Tyr6-Ile12*, with * denoting the partner molecule) (Figure 3d). The buried surface area of the dimeric interface of FnYqdH was 1613.40 Å^2^.

### 2.3. Substrate- and Cofactor-Binding Site of FnYqdH

As previously mentioned, both NADH and metal ions are required for the enzyme activity of FnYqdH. However, the detailed mechanism of the interaction of FnYqdH with cofactors remains unknown. In the crystal structure of FnYqdH-NAD-Co^2+^, an electron-density map corresponding to NADH and metal ions was clearly observed (Appendix A). NADH was located on the NBD, which was coordinated by the O atom of Pro75 (distance: 3.50 Å), OG atom of Thr143 (3.20 Å), OG atom of Ser152 (3.10 Å), and O atom of Val184 (2.70 Å). In addition, a water molecule acting as a bridge between NADH and two residues (Glue41 and Ser104) was coordinated by the OE1 and OE2 atoms of Glu41 (3.30 Å and 3.50 Å) and the OG atom of Ser104 (3.20 Å) (Figure 4a). The superposition of NADH-binding residues of FnYqdH-apo with those of FnYqdH-NAD-Co^2+^ showed that Pro75, Ser152, and Val184 do not undergo conformational changes, although Glu41, Ser104, and Thr143 were positioned differently in both FnYqdH-apo and FnYqdH-NAD-Co^2+^ (Figure 4a). Sequence alignment of FnYqdH with other BDHs from four different species, namely *C. beijerinkii*, *C. acetobutylicum*, *Bacillus megaterium*, and *Zymomonas mobilis*, showed Pro75, Ser104, and Thr143 to be highly conserved in all the selected species (Appendix A). However, Glu41, Ser152, and Val184 of FnYqdH were not conserved, suggesting that in FnYqdH, three residues—namely Pro75, Ser104, and Thr143—are significant for the NADH-binding of BDH. In addition, we found 12 residues (Gly40, Gly41, Ser42, Gly99, Ser100, Asp103, Thr139, Asn148, Asn150, Lys161, Val180, and Thr183) to be crucial for NADPH binding in the crystal structure of TmBDH (homolog protein of FnYqdH, described below). To identify the key residue involved in NAD(P)H binding, we produced nine FnYqdH mutant proteins (E41A, G44A, L46A, S104A, D107A, T143A, S152A, K165A, and T187A) in *Escherichia coli*, using the same procedure used for wild type FnYqdH (FnYqdH-WT), and performed enzyme activity studies. The results showed the enzyme activities of the L46A, S104A, D107A, T143A, S152A, and T187A mutants to be completely abolished. In contrast, the enzyme activities of the E41A, G44A, and K165A mutants were retained as compared with other FnYqdH mutants (Figure 4b), indicating that Leu46, Ser104, Asp107, Thr143, Ser152, and Thr187 are critical residues for NADH recognition.

To better understand the conformational change in proteins, we superposed the structures of FnYqdH-apo and FnYqdH-NAD-Co^2+^ and found both structures to be similar, with an r.m.s.d. of 0.375 Å (Appendix A). Notably, the crystal structures of FnYqdH-apo and FnYqdH-NAD-Co^2+^ revealed an open conformation, in which NBD and CD were formed at approximately 30° and 20°, respectively (Appendix A). Moreover, we found the β6-ƞ1 loop (L12) to be distanced from NADH in the crystal structure of FnYqdH-NAD-Co^2+^ compared to that of FnYqdH-apo (Appendix A). Next, we measured the distance between Ala103 on NBD and Gly196 on CD of FnYqdH-apo and FnYqdH-NAD-Co^2+^ and found the distance to be approximately 12.3 Å and 12.9 Å, respectively (Figure 5a,b). Upon NADH and metal binding to FnYqdH, NBD and CD revealed a narrower substrate-binding cleft compared to that of FnYqdH-apo.

The electron-density map corresponding to Co^2+^ was observed around the L22 loop and α7-helix. Co^2+^ was coordinated by the OE1 atom of Glu206 (2.80 Å), ND1 atom of His272 (3.10 Å), NE2 atom of His286 (3.80 Å), and a water molecule (2.50 Å), thereby establishing tetrahedral coordination (Figure 5c). In addition, a water molecule acted as a bridge between Co^2+^ and the ND1 atoms of His203 (3.50 Å) (Figure 5c). B-factor analysis revealed the temperature factors of Co^2+^ (117.00 Å^2^) to be higher than the average temperature factor of all atoms in the protein (59.10 Å^2^), thereby indicating that Co^2+^ loosely interacts with FnYqdH. To clarify, the catalytic metal-binding residues of FnYqdH and the Co^2+^-binding residues were substituted with alanine residues, and enzyme activity was measured. BDH activity of D199A, H203A, H272A, and H286A mutants was completely abolished, whereas the enzyme activity of the E206A mutant was retained partially as compared with other FnYqdH mutants (Figure 5d).

The Co^2+^-binding affinity of FnYqdH-WT and the mutant proteins was measured using a microscale thermophoresis (MST) assay. The results showed that the *K*_D_ of WT, D199A, E206A, and H286A mutants was 63.9 μM, 47.4 μM, 19.6 μM, and 46.4 μM, respectively (Figure 5e). In contrast, no Co^2+^ binding was detected in H203A and H272A mutants (Figure 5e). Based on the biochemical and structural analyses of metal-binding residues, we concluded that His203 and His272 of FnYqdH are key residues for metal ion binding. Although Asp199 and His286 are not involved in metal ion binding, and these two residues play a pivotal role in FnYqdH activity.

### 2.4. Comparison of FnYqdH with Structural Homolog Proteins

To understand the molecular function of FnYqdH, homologous structures of FnYqdH were searched. The results showed that FnYqdH has structural homology with TmBDH (Protein Data Bank [PDB] code: 1VLJ; sequence identity = 40%; Z-score = 46.4), which is complexed with NADPH and Fe^3+^ (unpublished data). The domain organization and overall fold of TmBDH exhibited a high similarity to those of FnYqdH, but the superposition of FnYqdH-NAD-Co^2+^ and TmBDH structures illustrated a conformational difference (Figure 6a). In addition, the surface structure of FnYqdH-NAD-Co^2+^ revealed open conformation between the NBD and CD domains (approximately 20°), whereas TmBDH showed closed conformation between the NBD and CD domains (Figure 6b). In TmBDH, NADPH was stabilized via hydrogen bonds with 12 residues (Gly40, Gly41, Ser42, Gly99, Ser100, Asp103, Thr139, Asn148, Asn150, Lys161, Val180, and Thr183) (Figure 6c). Sequence alignment showed Gly44, Ser104, Asp107, Thr143, Lys165, Val184, and Thr187 of FnYqdH to be highly conserved in both FnYqdH and TmBDH, whereas Gly41, Leu46, Ala103, Ser152, and Ser154 of FnYqdH were not conserved in TmBDH (Figure 6d). Structure comparison of FnYqdH and TmBDH showed that some conserved amino acid residues (Gly44, Ser104, Asp107, and Lys165 of FnYqdH) have different conformations (Figure 6c).

In the crystal of TmBDH, Fe^3+^ was coordinated with the OD1 atom of Asp195 (2.80 Å), NE2 atom of His199 (2.20 Å), NE2 atom of His268 (2.30 Å), and NE2 atom of His282 (2.20 Å), forming tetrahedral coordination (Figure 6e). This metal ion position of TmBDH differs from that of Co^2+^ in FnYqdH. In addition, Asp199, His203, Glu206, His272, and His286 of FnYqdH had different conformations when compared with TmBDH. In particular, Asp195 and His199 of TmBDH directly interacted with Fe^3+^, whereas His203 of FnYqdH relied on a water molecule for Co^2+^, and Glu206 of FnYqdH interacted with Co^2+^ instead of Asp199 (Figure 5c and Figure 6e). To understand the effect of different metal ions on the enzyme activity of metal-free TmBDH (the metal of TmBDH is removed by incubation with EDTA, a chelating agent), the enzyme activity was measured using a spectrophotometric assay for monitoring NADH consumption for five different metal ions (Mg^2+^, Co^2+^, Ni^2+^, Zn^2+^, and Fe^3+^). The highest enzyme activity of TmBDH was observed in the presence of Zn^2+^. The overall BDH activity trend was as follows: Co^2+^ > Ni^2+^ > Fe^3+^, with relative activities of 91.2%, 80.1%, and 23.8%, respectively. Accordingly, the enzyme activity of TmBDH was enhanced by the addition of Zn^2+^, Co^2+^, and Ni^2+^; however, Fe^3+^ did not significantly increase the activity, which was contrary to that in FnYqdH (Figure 6f). Moreover, we found that the NADH consumptions of TmBDH were 473 μM, 432 μM, and 381 μM in the presence of Zn^2+^, Co^2+^, and Ni^2+^, respectively (Figure 6f and Appendix A). Accordingly, based on our biochemical study, we determined that the crystal structure of Fe^3+^-bound TmBDH is not the most active conformation in the metal-binding site.

## 3. Discussion

Butanol has important industrial applications as a fuel additive and a raw chemical for the plastics industry [28,29] and as a grade extractant in the food and flavor industry [30]. BDH plays a key role in butanol biosynthesis by converting butanal to butanol, using NAD(P)H as a cofactor [31]. However, current research on BDH largely focuses on non-pathogenic bacteria, such as *C. acetobutylicum* BDH and *C. beijerinkii* BDH, while several reports focus on BDH within pathogenic bacteria. To investigate whether the enteropathogens *Clostridioides difficile* [32] and *Clostridium perfringens* [33] use the same butyrate metabolism mechanism as *F. nucleatum*, we examined their butyrate metabolism pathways using KEGG [34]. Through KEGG analysis, we found that in *C. difficile,* butyrate can be reduced to butanol during butanoate metabolism in two ways:first, via the 3-oxoacid CoA-transferase, aldehyde/alcohol dehydrogenase, and alcohol dehydrogenase pathway, and, second, via the butyrate kinase, phosphotransbutyrylase, aldehyde/alcohol dehydrogenase, and alcohol dehydrogenase pathway (Appendix A) [34]. Moreover, we found that butyrate can be reduced to butanol during butanoate metabolism via the butyrate kinase, phosphate butyryltransferase, aldehyde/alcohol dehydrogenase, and alcohol dehydrogenase pathway in *C. perfringens* (Appendix A) [34]. Therefore, *C. difficile* has the same butyrate metabolic pathway as *F. nucleatum*, whereas the conversion of butyrate to butyrate-CoA in *C. perfringens* involves a different enzyme.

In this study, we first characterized BDH from *F. nucleatum*, which is an emerging human bacterial pathogen, and determined the crystal structure of FnYqdH in its apo and complexed forms (with NADH and Co^2+^). We also characterized TmBDH, whose crystal structure had been deposited in the PDB by another research group (unpublished). The BDH family is classified into BDHA and BDHB based on the cofactors NADH and NADPH, respectively. Interestingly, FnYqdH exhibited BDH activity using both NADH and NADPH. Moreover, we found that the use of both NADH and NADPH as cofactors is not distinctive to FnYqdH. Yao et. al. found that the bifunctional alcohol–aldehyde dehydrogenase (AdhE) of *Thermoanaerobacter mathranii* showed a small amount of NADPH activity in addition to NADH activity [35]. Pei et. al. found that the AdhE of *Thermoanaerobacter ethanolicus* showed NADH aldehyde dehydrogenase activity and small amounts of NADPH alcohol dehydrogenase activity [36]. Biswas et. al. found that the Asp-494-Gly (D494G) mutation in AdhE of *Clostridium thermocellum* causes the enzyme to use both NADH and NADPH as cofactors, while WT prefers to use only NADH as a cofactor [37,38]. Further, the AdhE cofactor specificity is thought to be determined by the presence or absence of electrostatic repulsion and steric hindrance in the 2’-phosphate binding pocket of NADPH [38]. In the crystal structure of FnYqdH-NAD-Co^2+^, the 2’-phosphate of NADPH (aligned with TmBDH) is not sterically hindered in the binding pocket, which implies that NADPH can enter the binding pocket (Appendix A). Thus, FnYqdH exhibits bifunctional cofactor specificity.

FnYqdH has broad substrate specificity, showing the highest activity with heptanal and lowest with methanal. FnYqdH can convert aldehyde to alcohol by utilizing NAD(P)H with broad substrate specificity; however, it has no enzyme activity toward alcohols. Accordingly, the enzyme activity of FnYqdH differed from that of typical ADH, which catalyzes the oxidation of alcohols and reduction of aldehydes [26]. These findings indicate that the reductase activity of FnYqdH is useful for producing biorenewable fuels and chemicals, such as butanol, pentanol, hexanol, and heptanol. The in vitro biochemical study herein revealed that FnYqdH shows BDH activity in the presence of Co^2+^, Ni^2+^, Zn^2+^, and Fe^3+^, with Co^2+^ being associated with the highest activity. Interestingly, even though Fe^3+^ is bound to the crystal structure of TmBDH, TmBDH showed low activity with Fe^3+^ and high activity in the presence of Zn^2+^. Since our experiments on optimal metal ions for BDH activity were performed in vitro, further in vivo experiments on the preferred metals are warranted. Combining these biochemical experiments of FnYqdH and TmBDH suggests two useful experimental directions for the industrial application of BDH. First, since BDH activity changes in vitro depending on the type of metal, the target BDH for industrial application can increase productivity by screening and selecting metals with optimal activity. Second, the substrate specificity of BDHs is not limited to butanal. BDHs can be applied to convert various aldehydes into alcohols, highlighting the potential for the industrial production of butanol and other alcohols.

Crystal structures of FnYqdH-apo and FnYqdH-NAD-Co^2+^ showed an open conformation between NBD and CD, with a broad substrate-binding cleft of more than 12 Å present between the two domains. The distance between the two domains was reduced by approximately 1 Å upon binding NADH and Co^2+^, indicating that the cofactor did not significantly affect the conformation between the two domains of FnYqdH. For the BDH activity of FnYqdH, it is essential that NAD and metal ions involved in the activity are in close proximity. Indeed, for the two cofactors to be close, the two domains must be close. Accordingly, in the crystal structures of FnYqdH determined in this experiment, the open conformation corresponded to the pre-substrate binding state and not the close conformation of the active state. Meanwhile, FnYqdH preferred aldehydes with an aliphatic chain length of approximately 3–8, thereby indicating that the aliphatic chain of aldehyde can interact with FnYqdH. As a result, we expect that the open conformation between the NBD and CD domains of FnYqdH will change to a closed conformation when FnYqdH binds to a substrate or undergoes a BDH reaction. In contrast, the crystal structure of TmBDH showed a closed conformation, maintaining a close distance to the nucleotide and metal compared to FnYqdH; hence, it could be determined whether it was very close to the active state. To understand the substrate recognition and catalytic mechanisms, the crystal structure of FnYqdH complexed with cofactors and substrate is required.

Moreover, based on structural analyses, we identified key residues involved in NADH and metal binding through a comprehensive study, including mutagenesis, biochemical assay for BDH activity, and MST assays. The results of these mutagenesis and biochemical assays provide a deeper understanding of the molecular properties of the BDH family. However, the substrate-binding site remains unknown. The ADH of *C. acetobutylicum* (CaADH) consists of an N- and a C-terminal domain, which form two clefts. NADH occupies one cleft, and the other forms a substrate-binding chamber [39]. Structural comparison of FnYqdH-NAD-Co^2+^ and a model structure of CaADH, *Geobacillus thermoglucosidasius* ADH (PDB code: 3ZDR, r.m.s.d.: 2.503 Å), showed that FnYqdH has a substrate-binding chamber located in a similar position (Appendix A). In addition, Rellos et. al. found that Ala161 is at the ADH of *Z. mobilis* (ZmADH) substrate-binding site, and its conversion to Val or Ile also allows butanol oxidization [40]. Sequence alignment of FnYqdH and ZmADH showed that Ala161 is not highly conserved, but in FnYqdH, it is converted to Val (Val167), which is a hydrophobic amino acid similar to Ala (Appendix A). Structural comparison of FnYqdH-NAD-Co^2+^ and ZmADH (PDB code: 3OX4, r.m.s.d.: 1.138 Å) showed that Val163 (FnYqdH) and Ala161 (ZmADH) are in similar positions (Appendix A). Sequence alignment of FnYqdH and other ADHs showed that most of the amino acids at this site are hydrophobic (Appendix A). Therefore, we determined that these results imply the direct involvement of Val167 of FnYqdH in substrate binding. Of course, further experiments should be carried out to verify this hypothesis.

In nature, various types of dehydrogenases are utilized in extended proton relay systems. Among them, the molecular mechanism of *Thermoanaerobacter brockii* ADH (TbADH) has been well established [41]. In TbADH, the binding of the alcohol substrate is considered to form a penta-coordinated zinc complex [41]. In the catalytic cycle, two supposed penta-coordinate intermediates are formed. Specifically, the tetra-coordinated zinc ion species is maintained when NADP^+^ is bound to TbADH, and the first transient penta-coordinated zinc ion complex is formed by adding a water molecule [41]. The penta-coordinated zinc ion species is then reconverted to a tetra-coordinated zinc ion complex due to the dissociation of ligated Glu60 residue. When the alcohol substrate is bound, a penta-coordinated zinc ion complex is formed, which corresponds to the second transient species. When the water molecule and product dissociate, the original tetra-coordinated zinc ion species are regenerated by re-ligating Glu60, and the catalytic cycle is complete, returning the enzyme to its resting state [41]. By active site analysis of FnYqdH, we found Glu206 to be located nearby metal ions and TbADH; this residue was highly conserved in other BDHs (Appendix A). Site-directed mutagenesis and MST assay revealed that the enzyme activity of FnYqdH-E206A was reduced, although metal-binding affinity persisted. The results suggested that Glu206 of BDH is involved in the catalytic process and interacts weakly with metal ions.

Based on previous reports and our results, we proposed an alternative model for a putative proton relay of FnYqdH. When NADH and a water molecule are bound to FnYqdH, a transient tetra-coordinated Co^2+^ complex is formed. The binding of the incoming aldehyde substrate forms a penta-coordinated Co^2+^ complex. The latter is then transformed into a tetra-coordinated Co^2+^ complex by dissociating the ligated Glu206 residue. The catalytic cycle is envisaged to restart by binding the water molecule and NADH (Figure 7). Based on the results above, we considered that further experimental investigations would be required to ensure whether the systems would indeed apply to other BDHs.

## 4. Materials and Methods

### 4.1. Cloning, Expression, and Purification

Genomic DNA from *F. nucleatum* was used as a template to amplify the yqdH gene (GenBank: AAL95608.1) by a conventional polymerase chain reaction (PCR). The PCR fragment was digested at the NcoI and XhoI restriction sites and cloned into the pPRO-EX-HTA vector. Expression and purification of FnYqdH were performed as previously described [42]. Site-directed mutagenesis was performed by two subsequent PCR reactions using the primers shown in Appendix A. Mutant proteins were expressed and purified using the same procedure as the FnYqdH-WT.

### 4.2. Enzyme Activity Assays

Enzyme activity of FnYqdH was determined using butanal as a substrate to measure NAD(P)H consumption. The reaction mixture contained 10 mM Tris-HCl, pH 6.0, 0.5 mM NADH or NADPH, 50 mM butanal, 2 mM CoSO_4_, and an enzyme in a reaction volume of 100 μL. The reaction mixture was incubated for 10 min at 35 °C and then quickly placed in an ice bath for 5 min, after which the reaction was terminated. Enzyme activities were then assayed by monitoring NAD(P)H consumption using a microplate spectrophotometer at 340 nm. All experimental data are averages of at least triplicates.

### 4.3. Standard Curve of NADH and NADPH

The NAD(P)H concentration curve was measured using a microplate spectrophotometer at 340 nm. Serial dilutions of NAD(P)H ranging from 0–700 μM were made using 10 mM Tris-HCl, pH 6.0, 2 mM CoSO_4_, and 50 mM butanal. All experimental data are averages of at least triplicates.

### 4.4. Metal Substitution

Purified FnYqdH and TmBDH proteins were each diluted two-fold with chelating buffer containing 20 mM Tris-HCl, pH 8.0, 150 mM NaCl, and 10 mM EDTA and incubated at 25 °C for 1 h. The protein solution was further dialyzed using a buffer containing 20 mM Tris-HCl, pH 8.0, and 150 mM NaCl at 25 °C overnight. Further, metal-free FnYqdH and TmBDH proteins were incubated with 2 mM metal ions (ZnCl_2_, MgCl_2_, CoSO_4_, FeCl_3_, and NiCl_2_).

### 4.5. Substrate Spectrum

Substrate specificity of FnYqdH was determined. The activity of FnYqdH in reducing methanal, ethanal, propanal, butanal, pentanal, hexanal, heptanal, octanal, nonanal, and decanal was detected by the standard method. The reaction system (100 μL) usually consisted of the corresponding substrate, as mentioned above, in 10 mM Tris-HCl, pH 6.0, 0.5 mM NADH, 2 mM CoSO_4_, and an appropriate amount of enzyme.

The activity of FnYqdH in the oxidation of methanol, ethanol, propanol, butanol, pentanol, hexanol, heptanol, octanol, nonanol, and decanol using NAD^+^ as a coenzyme was used to measure NADH production. The reaction system (100 μL) usually consisted of the corresponding substrate, as mentioned above, in 10 mM Tris-HCl, pH 6.0, 0.5 mM NAD^+^, 2 mM CoSO_4_, and an appropriate amount of enzyme. It was then assayed by monitoring NADH production using a microplate spectrophotometer at 340 nm. All experimental data are averages of at least triplicates.

The double reciprocal plot was applied to obtain the kinetic parameters of FnYqdH in the presence of NADH (0.1–0.5 mM) or NADPH (0.1–1 mM).

### 4.6. Crystallization and Data Collection

Crystallization and preliminary X-ray diffraction analysis of FnYqdH-apo have been published previously [42]. High-quality FnYqdH crystals were incubated for 1 min in a solution consisting of 25% (*v*/*v*) glycerol. X-ray diffraction data were collected with a Beamline 5C from the Pohang Light Source (PLS, Pohang, Republic of Korea) at −173 °C. Diffraction images were recorded with an ADSC Q315 CCD detector and processed with the HKL2000 program. For FnYqdH complexed with NADH and Co^2+^, a crystal of FnYqdH was soaked in a reservoir solution consisting of 25% (*v*/*v*) glycerol, 10 mM NADH, and 2 mM Co^2+^ and incubated for 1 min. X-ray diffraction data were collected at beamline BL17B of the Shanghai Synchrotron Radiation Facility (SSRF, Shanghai, China) at −173 °C. Diffraction images were recorded with a Rayonix MX300 detector (Rayonix, LLC, Evanston, IL, USA) and processed with the HKL2000 program [43,44].

### 4.7. Structure Determination and Refinement

The crystal structure of FnYqdH was determined by the molecular replacement (MR) method using the Phenix program [45,46]. The crystal structure of TmBDH (PDB code: 1LVJ) was used as a search template model. Model building and refinement were conducted with the Crystallographic Object-Oriented Toolkit (COOT) [47] and Phenix [45]. The structure of FnYqdH-NAD-Co^2+^ was determined by an MR method, using the native structure of FnYqdH as the search model. Ligands and water molecules were added by manually inspecting a 2Fo-Fc electron-density map as a guide in COOT [47]. All figure structures were generated using the program PyMOL [48]. The geometry of the final models was validated with MolProbity [49]; homologous structures of FnYqdH were searched using the DaliLite server [50].

### 4.8. Size-Exclusion Chromatography

To determine the molecular weight and the oligomeric state of FnYqdH, purified FnYqdH (500 μL of 1 mg/mL) was loaded onto a Superdex 200 10/300 GL column (GE Healthcare, Chicago, IL, USA), and the protein was eluted using a buffer containing 20 mM Tris-HCl, pH 8.0, 150 mM NaCl, and 2 mM β-mercaptoethanol (BME), at a flow rate of 0.5 mL/min.

### 4.9. Labeling and MST Measurements

The binding constant of FnYqdH, with respect to the substrate, was measured using MST. Further, 400 nM of purified FnYqdH and mutant proteins and 100 nM dye were diluted in assay buffer (20 mM HEPES, pH 8.0, and 0.5% polysorbate 20). The protein and the dye were mixed and incubated at 25 °C for 30 min. After incubation, the supernatant was collected by centrifugation at 13,000 rpm for 15 min at 4 °C. Increasing concentrations (380 nM to 12.5 mM) of CoSO_4_ in the assay buffer were mixed with constant amounts of labeled FnYqdH or mutant proteins. Samples were loaded into standard capillaries (Monolith NT.115), and all measurements were performed at 40% LED power and medium MST power, in duplicates. Data evaluation and *K*_D_ value determination were performed using NT analysis software (Nano Temper Technologies, Munich, Germany) [51].

## 5. Conclusions

We comprehensively analyzed the molecular function of FnYqdH using biochemical studies, structure analyses, and mutagenesis. Based on our and previously reported results, we proposed a putative proton relay system for the BDH activity of FnYqdH. Our results expand knowledge of the BDH family, thereby providing insight into the future application of BDH in industry.

## Figures and Tables

**Figure 1 ijms-24-02994-f001:**
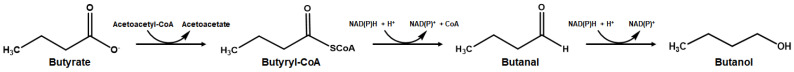
Butanol synthesis in *F. nucleatum*.

**Figure 2 ijms-24-02994-f002:**
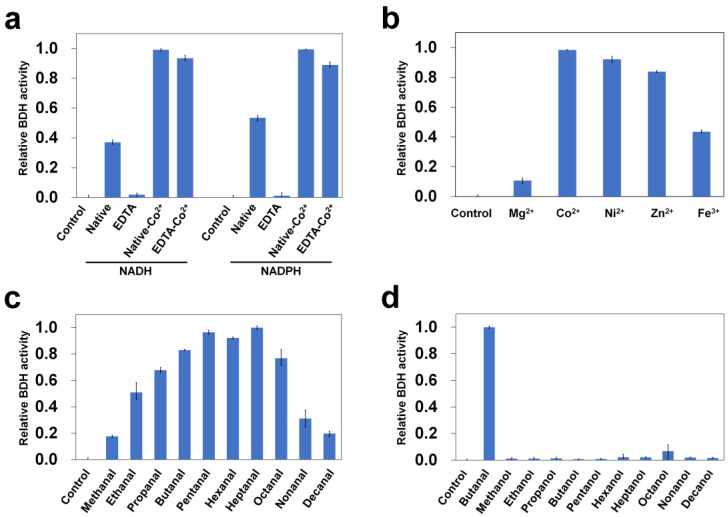
Butanol dehydrogenase activity of FnYqdH. (**a**) Effect of cofactors on enzyme activity of FnYqdH. (**b**) Effect of metal ions on enzyme activity of FnYqdH. Measurement of enzyme activity of FnYqdH toward the (**c**) aldehydes and (**d**) alcohols (the activity of butanal was defined as 100%). The graph represents the mean of three independent experiments, and error bars indicate the standard deviation.

**Figure 3 ijms-24-02994-f003:**
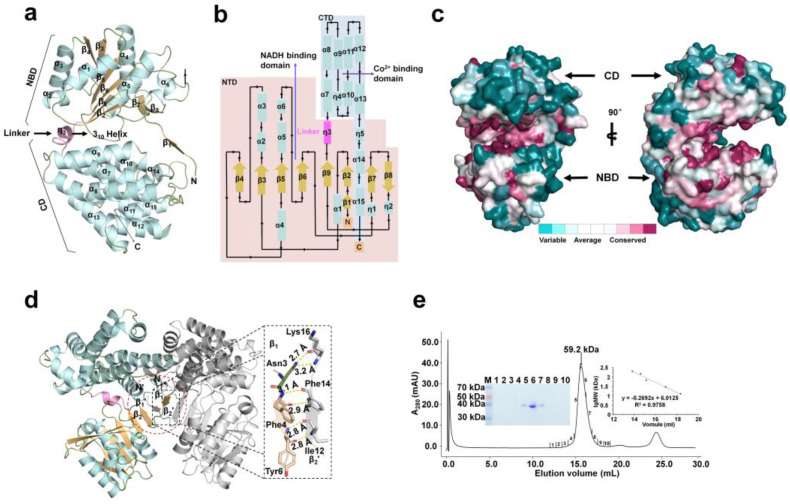
Crystal structure of FnYqdH. (**a**) Cartoon representation of monomer structure of FnYqdH consisting of the NBD, CD, and linker. (**b**) Topology diagram of FnYqdH. The α-helices (cyan) and β-strands (yellow) are indicated by rectangles and arrows, respectively. The smaller numbers indicate the beginning and ending residue number of each secondary structural element. The NTD is shadowed by light orange boxes. The CTD of FnYqdH is marked by a light cyan box. The linker is marked by a pink box. (**c**) Surface conservation of FnYqdH. The substrate-binding cleft between NBD and CD shows high conservation, whereas other regions are not conserved. (**d**) Dimer formation of FnYqdH. The hydrogen bonds between β1 and β2′ strands stabilize the dimeric interface of FnYqdH. (**e**) Profile of size-exclusion chromatography of FnYqdH revealed the dimer state in solution.

**Figure 4 ijms-24-02994-f004:**
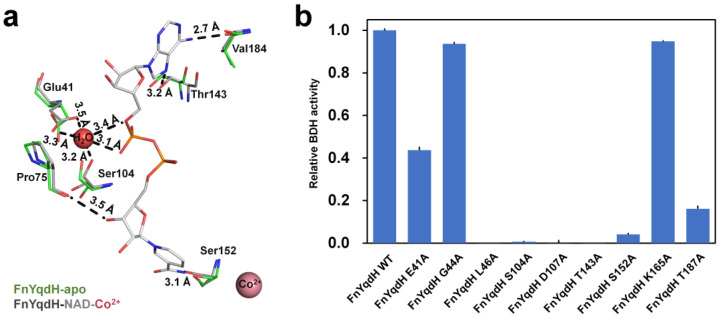
NADH-binding site in FnYqdH. (**a**) Superposition of the NADH-binding site in FnYqdH (green) and FnYqdH-NAD-Co^2+^ (gray). (**b**) BDH activity of FnYqdH mutants for NADH-binding residues (E41A, G44A, L46A, S104A, D107A, T143A, S152A, K165A, and T187A). The activity of wild type FnYqdH (FnYqdH-WT) was defined as 100%.

**Figure 5 ijms-24-02994-f005:**
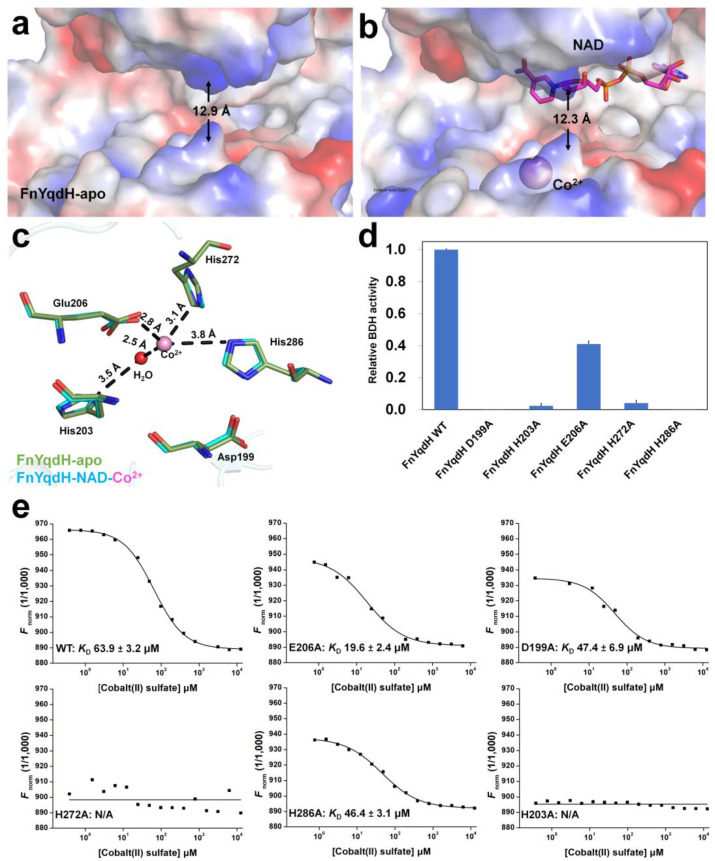
The metal-binding site of FnYqdH. The surface structure of substrate-binding cleft of (**a**) FnYqdH-apo and (**b**) FnYqdH-NAD-Co^2+^. (**c**) Superposition of the Co^2+^-binding site of FnYqdH (green) and FnYqdH-NAD-Co^2+^ (cyan). (**d**) BDH activity of FnYqdH with mutated metal-binding residues (FnYqdH-D199A, FnYqdH-H203A, FnYqdH-E206A, FnYqdH-H272A, and FnYqdH-H286A). The activity of FnYqdH-WT was defined as 100%. (**e**) MST assays of the Co^2+^-binding affinity of FnYqdH-WT and its mutants (FnYqdH-D199A, FnYqdH-H203A, FnYqdH-E206A, FnYqdH-H272A, and FnYqdH-H286A). No Co^2+^-binding affinity of FnYqdH-H203A and FnYqdH-H272A was detected. *K*_D_ values 63.9 μM, 19.6 μM, 47.4 μM, and 46.4 μM for Co^2+^, FnYqdH-WT, FnYqdH-E206A, FnYqdH-D199A, and FnYqdH-H286A, respectively.

**Figure 6 ijms-24-02994-f006:**
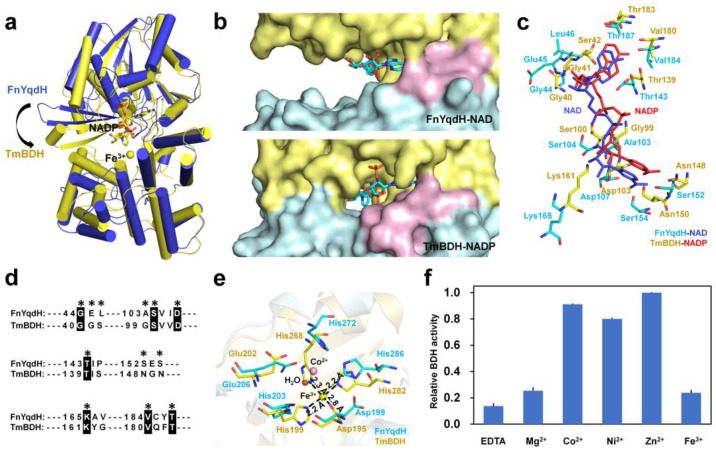
Structural and functional comparison of FnYqdH with TmBDH. (**a**) Superposition of monomeric structures of FnYqdH (blue) and TmBDH (yellow). (**b**) Comparison of the surface structures of FnYqdH-NAD-Co^2+^ and TmBDH. (**c**) Superposition of NAD(P)H-binding sites of FnYqdH-NAD-Co^2+^ (cyan) and TmBDH (yellow). NADH and NADPH are colored dark blue and red, respectively. (**d**) Partial sequence alignment of FnYqdH and TmBDH. The NADPH-binding site of TmBDH is indicated by an *. A black background highlights conserved residues. (**e**) Superposition of the metal ion-binding sites of FnYqdH-NAD-Co^2+^ (cyan) and TmBDH (yellow). Co^2+^, Fe^3+^, and water molecules are colored pink, red, and yellow, respectively. The dark dashed lines indicate interactions between Fe^3+^ and amino acids in TmBDH. (**f**) Effect of metal ions on the enzymatic activity of TmBDH.

**Figure 7 ijms-24-02994-f007:**
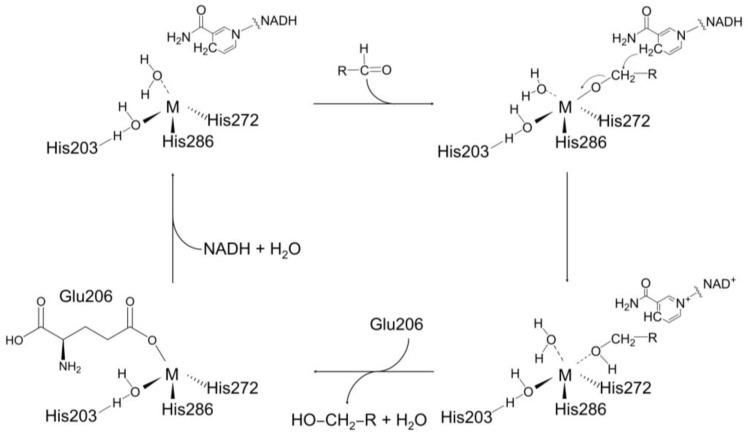
Proposed mechanism of BDH activity of FnYqdH. A tentative reaction mechanism involves extended proton relay systems.

**Table 1 ijms-24-02994-t001:** Crystallographic data and refinement statistics.

Data Collection	FnYqdH	FnYqdH-NAD-Co^2+^
Beamline	Beamline 5C at PLS	Beamline 17B at SSRF
Resolution range (Å)	37.78–1.98 (2.05–1.98)	35.53–2.72 (2.82–2.72)
Space group	I222	I222
Total/unique reflections	39065 (3723)	14920 (1334)
a, b, c (Å)	64.77, 78.85, 215.22	64.426, 79.351, 212.932
α, β, γ (^o^)	90.00, 90.00, 90.00	90.00, 90.00, 90.00
CC1/2	0.975 (0.883)	0.929 (0.531)
CC*	0.994 (0.968)	0.981 (0.833)
Completeness (%)	99.58 (97.21)	98.64 (87.87)
Multiplicity	6.6 (6.0)	12.4 (10.3)
Average I/σ(I)	15.26 (1.8)	11.88 (1.6)
**Refinement**		
R_work_/R_free_ (%)	17.7/20.9	20.5/26.0
No. of non-hydrogen atoms	3342	3121
Protein residues	384	384
B-factor (Å^2^)	29.01	59.10
Protein	28.199	58.71
NADH		94.57
Co^2+^		117.00
Water	36.97	50.00
R.m.s.d. from ideal		
RMS (Bond)	0.008	0.009
RMS (Angles)	0.81	1.61
Ramachandran plot (%)		
Favored regions	98.69	93.72
Allowed regions	1.31	6.02
Disallowed regions	0.00	0.26
PDB code	6L1K	8I29

Values in parentheses are for the outermost shell.

## Data Availability

Coordinates and structure factors of FnYqdH and FnYqdH-NAD-Co^2+^ were deposited in RCSB PDB with accession codes 6L1K and 8I29, respectively.

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
