# Peer review of "Structural and Biochemical Analyses of the Butanol Dehydrogenase from *Fusobacterium nucleatum"

_ijms, 2023, doi:10.3390/ijms24032994_

Round 1

Reviewer 1 Report

The article is interesting and original and deserves to be published.

However, I have the following questions about it:

1)      Has the same mechanism of pathogenesis, i.e. production of butanol from butyrate, been described in other pathogenic microorganisms with the same capabilities, e.g. C. difficile or C.perfringens? If so, it would deserve to be added to the introduction and discussion.

2)      Why were different divalent cations tested to increase the activity of butanol dehydrogenase, along with trivalent iron? Why was the divalent iron cation not tested? What were the initial assumptions?

3)      The use of both NADH and NADPH as cofactors for alcohol/aldehyde dehydrogenase is not a specialty of the enzyme studied. The same has been described for various solventogenic non-pathogenic clostridial species. This should be added to the discussion.

Author Response

Reviewer1

Major comments

Has the same mechanism of pathogenesis, i.e. production of butanol from butyrate, been described in other pathogenic microorganisms with the same capabilities, e.g. C. difficile or C.perfringens? If so, it would deserve to be added to the introduction and discussion.

Response: Thank you for your valuable question. Accordingly, we discussed the butyrate metabolism pathway of C. difficileand and C.perfringens in the discussion section.

Added (line 328-340)

To investigate whether the enteropathogens Clostridioides difficile [32] and Clostridium perfringens [33] use the same butyrate metabolism mechanism as that in F. nucleatum, we examined their butyrate metabolism pathways using KEGG [34]. Through KEGG analysis, we found that in C. difficile, butyrate can be reduced to butanol during butanoate metabolism in two ways, first via the 3-oxoacid CoA-transferase, aldehyde/alcohol dehydrogenase, and alcohol dehydrogenase pathway, and second via the butyrate kinase, phosphotransbutyrylase, aldehyde/alcohol dehydrogenase, and alcohol dehydrogenase pathway (Figure S9) [34].  Moreover, we found that butyrate can be reduced to butanol during butanoate metabolism via the butyrate kinase, phosphate butyryltransferase, aldehyde/alcohol dehydrogenase, and alcohol dehydrogenase pathway in C. perfringens (Figure S10) [34]. Therefore, C. difficile has the same butyrate metabolic pathway as F. nucleatum, whereas the conversion of butyrate to butyrate-CoA in C. perfringens involves a different enzyme.

Added (Supplementary Figure S9 and S10)

Figure S9. Butanoate metabolism of Clostridioides difficile.

Figure S10. Butanoate metabolism of Clostridium perfringens.

Why were different divalent cations tested to increase the activity of butanol dehydrogenase, along with trivalent iron? Why was the divalent iron cation not tested? What were the initial assumptions?

■ Response: Thank you for the important questions. Accordingly, we included the reason why the divalent iron cation not tested.

Added (line 114-116)

Meanwhile, since Fe2+ is easily oxidized to Fe3+ at pH 6.0 [25], butanol dehydrogenase activity was monitored only in the presence of Fe3+ to avoid possible experimental errors when Fe2+ was used.

The use of both NADH and NADPH as cofactors for alcohol/aldehyde dehydrogenase is not a specialty of the enzyme studied. The same has been described for various solventogenic non-pathogenic clostridial species. This should be added to the discussion.

■ Response: Thank you for the valuable suggestion. Accordingly, we included other ADH cofactor specificities in the discussion section.

Revised (line 347-360)

Moreover, we found that the use of both NADH and NADPH as cofactors is not distinctive of FnYqdH. Yao et. al. found that the bifunctional alcohol–aldehyde dehydrogenase (AdhE) of Thermoanaerobacter mathranii showed a small amount of NADPH activity in addition to NADH activity [35]. Pei et. al. found that the AdhE of Thermoanaerobacter ethanolicus showed NADH aldehyde dehydrogenase activity and small amounts of NADPH alcohol dehydrogenase activity [36]. Biswas et. al. found that Asp-494-Gly (D494G) mutation in AdhE of Clostridium thermocellum causes the enzyme to use both NADH and NADPH as cofactors, while WT prefers to use only NADH as a cofactor [37, 38]. Further, AdhE cofactor specificity is thought to be determined by the presence or absence of electrostatic repulsion and steric hindrance in the 2'-phosphate binding pocket of NADPH [38]. In the crystal structure of FnYqdH-NAD-Co2+, the 2'-phosphate of NADPH (aligned with TmBDH) is not sterically hindered in the binding pocket, which implies that NADPH can enter the binding pocket (Figure S11). Thus, FnYqdH exhibits bifunctional cofactor specificity.

Added (Supplementary Figure S11)

Figure S11. (a) The surface structure of the NAD(P)H-binding pocket of FnYqdH-NAD-Co2+, NADH is colored blue and NADPH is colored green. (b) Spatial distribution of NAD(P)H-binding pocket and NAD(P)H.

Reviewer 2 Report

The paper reported the functional characterization and structures of BHD from Fusobacterium nucleatum. They found the FnBHD requires metal cations as cofactors, and can utilize both NADH and NADPH as cofactors in its catalytic reaction. I have some questions based on the reported results:

Major:

1. In the functional characterization section, all the assays were performed in single concentration and time points, and the results are presented as relative activity. I wonder if the single-point data is enough and precise to present the real substrate, metal ion, and cofactor selectivity here. If the substrates were used up more than 10% of the total amount, the readout number will not precisely represent the initial rate of the reaction. Therefore, I suggest the authors include the raw readouts in the supplementary file so that we can know the consumption percentage of the substrates.

2. With a similar reason as question 1, I suggest the author compare the NADH/NADPH preference with a Km/Vmax curve rather than single-point data. The Km value will be more convincing than the relative activity to compare the substrate preference. 

3. The manuscript reports the structure of FnBHD in apo and in complex with NADH/Co2+. However, the substrate binding site is still unknown. Are there any hints in current and previous studies that indicate the potential binding site of butanol?

Minor:

1. The authors need to revise the language to make the phrases more precise and clear.

2. I suggest the author include the topology of the protein in figure3 and highlight the relative position of cation and NADH in the overall structure of FnBDH. 

3. I suggest the author refine the model of the cation and NADH-bound structure to remove the Ramachandran outliners. Considering the high resolution of the data, it should be easy to get a model with zero Ramachandran outliners.

Author Response

Reviewer 2

Major comments

In the functional characterization section, all the assays were performed in single concentration and time points, and the results are presented as relative activity. I wonder if the single-point data is enough and precise to present the real substrate, metal ion, and cofactor selectivity here. If the substrates were used up more than 10% of the total amount, the readout number will not precisely represent the initial rate of the reaction. Therefore, I suggest the authors include the raw readouts in the supplementary file so that we can know the consumption percentage of the substrates.

■ Response: Thank you for your constructive suggestion. Accordingly, we included the raw readouts in a supplementary file (excel file: Table S2. Raw readout for characterization of FnYqdH.).

With a similar reason as question 1, I suggest the author compare the NADH/NADPH preference with a Km/Vmax curve rather than single-point data. The Km value will be more convincing than the relative activity to compare the substrate preference.

■ Response: Thank you for the valuable suggestion. Accordingly, we compared the NADH/NADPH preference with a Km/Vmax curve.

Added (line 107-112)

In the presence of 4.5 μM FnYqdH, the consumption of NADH and NADPH was 417 μM and 357 μM, respectively, (Figure 2a and Figure S1), indicating the ability of FnYqdH to use both NADH and NADPH as cofactors. Furthermore, the apparent Km and Vmax values were measured with a purified protein. The results showed that NADH is preferred over NADPH in terms of FnYqdH activity (Figure S2).

Added (Supplementary Figure S2)

Figure S2. The double reciprocal plot was used to calculate Vmax and Km of the enzyme in the presence of NADH and NADPH. Data represent average ± SD of three different samples.

The manuscript reports the structure of FnBHD in apo and in complex with NADH/Co2+. However, the substrate binding site is still unknown. Are there any hints in current and previous studies that indicate the potential binding site of butanol?

■ Response: Thank you for the important question. Accordingly, we compared the potential binding site analyses of butanol in the discussion section.

Revised (line 403-419)

However, the substrate-binding site remains unknown. The ADH of C. acetobutylicum (CaADH) consists of an N- and a C-terminal domain, which form two clefts. NADH occupies one cleft and the other forms a substrate-binding chamber [39]. Structural comparison of FnYqdH-NAD-Co2+ and a model structure of CaADH, Geobacillus thermoglucosidasius ADH (PDB code: 3ZDR, r.m.s.d.: 2.503 Å), showed that FnYqdH has a substrate-binding chamber located in a similar position (Figure S12). In addition, Rellos et. al. found that Ala161 is at the ADH of Z. mobilis (ZmADH) substrate-binding site, and its conversion to Val or Ile also allows butanol oxidization [40]. Sequence alignment of FnYqdH and ZmADH showed that Ala161 is not highly conserved, but in FnYqdH, it is converted to Val (Val167), which is a hydrophobic amino acid similar to Ala (Figure S12). Structural comparison of FnYqdH-NAD-Co2+ and ZmADH (PDB code: 3OX4, r.m.s.d.: 1.138 Å) showed that Val163 (FnYqdH) and Ala161 (ZmADH) are in similar positions (Figure S12). Sequence alignment of FnYqdH and other ADHs showed that most of the amino acids at this site are hydrophobic (Figure S5 and 8). Therefore, we consider that these results imply the direct involvement of Val167 of FnYqdH in substrate binding. Of course, further experiments should be tested to verify this hypothesis.

Added (Supplementary Figure S12)

Figure S12. Comparison of FnYqdH-NAD-Co2+ and other structures. (a) Schematic representation of FnYqdH-NAD-Co2+ (colored blue) and CaADH (model structure, colored grey). (b) Partial sequence alignment of FnYqdH and ZmADH. The substrate-binding site of ZmBDH is indicated by a red *. (c)Comparison of FnYqdH-NAD-Co2+ (colored marine [blue]) and ZmADH (colored green).

Minor

The authors need to revise the language to make the phrases more precise and clear.

■ Response: Thank you, according to reviewer’s suggestion, our manuscript was edited by a professional English editing company.

I suggest the author include the topology of the protein in figure3 and highlight the relative position of cation and NADH in the overall structure of FnBDH.

■ Response: Thank you for the helpful suggestion. Accordingly, we included the topology of the protein in Figure3 and highlighted the relative position of the cation and NADH.

Revised (Figure 3b)

Revised (line 187-191)

Figure 3. (b) Topology diagram of FnYqdH. The α-helices (cyan) and β-strands (yellow) are indicated by rectangles and arrows, respectively. The smaller numbers indicate the beginning and ending residue number of each secondary structural element. The NTD is shadowed by light orange boxes. The CTD of FnYqdH is marked by a light cyan box. The linker is marked by pink box.

I suggest the author refine the model of the cation and NADH-bound structure to remove the Ramachandran outliners. Considering the high resolution of the data, it should be easy to get a model with zero Ramachandran outliners.

■ Response: Thank you for the valuable suggestion. Accordingly, we revised the NADH-bound structure. The revised coordinate of NADH-bound FnYqdH was deposited in PDB under accession code 8I29. Table 1 was revised using the new coordinate file.

Round 2

Reviewer 2 Report

The author provides the raw data and edited the figures based on my suggestions. I have no more questions about their work.